# A fibrin enhanced thrombosis model for medical devices operating at low shear regimes or large surface areas

**Rodrigo Méndez Rojano** [1]*, **Angela Lai**[2], **Mansur Zhussupbekov** [1], **Greg W. Burgreen**[3], **Keith Cook** [2], **James F. Antaki**[1]

**1** Meinig School of Biomedical Engineering, Cornell University, Ithaca, New York, United States of America, **2** Department of Biomedical Engineering, Carnegie Mellon University, Pittsburgh, Pennsylvania, United States of America, **3** Center for Advanced Vehicular Systems, Mississippi State University, Starkville, Mississippi, United States of America

* rm2235@cornell.edu

**Data Availability Statement:** The simulation solutions and scripts to reproduce the plots are shared in https://doi.org/10.5281/zenodo.6603340. The numerical solutions can be visualized using the

## Abstract

Over the past decade, much of the development of computational models of device-related thrombosis has focused on platelet activity. While those models have been successful in predicting thrombus formation in medical devices operating at high shear rates (> 5000 s⁻¹), they cannot be directly applied to low-shear devices, such as blood oxygenators and catheters, where emerging information suggest that fibrin formation is the predominant mechanism of clotting and platelet activity plays a secondary role. In the current work, we augment an existing platelet-based model of thrombosis with a partial model of the coagulation cascade that includes contact activation of factor XII and fibrin production. To calibrate the model, we simulate a backward-facing-step flow channel that has been extensively characterized in-vitro. Next, we perform blood perfusion experiments through a microfluidic chamber mimicking a hollow fiber membrane oxygenator and validate the model against these observations. The simulation results closely match the time evolution of the thrombus height and length in the backward-facing-step experiment. Application of the model to the microfluidic hollow fiber bundle chamber capture both gross features such as the increasing clotting trend towards the outlet of the chamber, as well as finer local features such as the structure of fibrin around individual hollow fibers. Our results are in line with recent findings that suggest fibrin production, through contact activation of factor XII, drives the thrombus formation in medical devices operating at low shear rates with large surface area to volume ratios.

## Author summary

Patients treated with blood-contacting medical devices suffer from clotting complications. Over the past decades, a great effort has been made to develop computational tools to predict and prevent clot formation in these devices. However, most models have focused on platelet activity and neglected other important parts of the problem such as the coagulation cascade reactions that lead to fibrin formation. In the current work, we incorporate

open-source tool ParaView https://www.paraview.org/. The fibrin augmented thrombosis model can be downloaded from https://github.com/rodrigomrxvi/FibrinModel.git. The thrombosis code runs on OpenFOAM v6 which can be obtained from https://openfoam.org/version/6/.

**Funding:** J. F. A. received funding from National Institutes of Health (grants NIH R01 HL089456 and NIH R01 HL086918). The funders had no role in study design, data collection and analysis, decision to publish, or preparation of the manuscript.

**Competing interests:** The authors have declared that no competing interests exist.

this missing element into a well-established and validated model for platelet activity. We then use this novel approach to predict thrombus formation in two experimental configurations. Our results confirm that to accurately predict the clotting process in devices where surface area to volume ratios are large and flow velocity and shear stresses remain low, coagulation reactions and subsequent fibrin formation must be considered. This new model could have great implications for the design and optimization of medical devices such as blood oxygenators. In the long term, the model could evolve into a functional tool to inform anticoagulation therapies for these patients.

## Introduction

Hemostasis is the ensemble of biochemical and biophysical processes that control blood clotting when a blood vessel is damaged. These processes work in a delicate balance to stop bleeding yet prevent excessive thrombus formation (thrombosis) [1]. Medical devices that contact blood disrupt the physiological hemostatic balance leading to thrombotic or hemorrhagic complications. This may occur in two ways. First, platelets are susceptible to supra-physiological shear stresses induced by blood flow in devices such as prosthetic heart valves (PHV) or ventricular assist devices (VAD) [2–4]. When stressed, platelets become activated releasing platelet agonists that amplify synergistic activity in a positive feed-back loop. Second, when blood is in contact with foreign bodies, proteins such as fibrinogen, factor XII, and albumin adsorb to the surface. Subsequently, factor XII becomes activated by surface contact triggering a series of enzymatic reactions known as the intrinsic coagulation cascade [5]. One product of the coagulation cascade is fibrin monomers, which associate to form protofibrils to form a fibrotic mesh that traps platelet aggregates and other blood cells [6]. These two events can jointly or independently lead to thrombus formation and influence the composition of the thrombus. For example, in low shear stress environments such as found in blood oxygenators and associated circuits, the level of platelet activation is low, and most thrombi are fibrin-rich red clots [7]. On the other hand, the formation of platelet-rich white thrombi has been linked to regions of high shear stress within PHV and VAD [8, 9]. In devices operating at low shear rates or with high surface area to volume ratios, selective inhibition of the intrinsic branch of the coagulation cascade has been able to dramatically reduce clot formation [10–12], suggesting that this fibrin-dominated pathway is the primary driving force of clot formation in such devices.

Computational models have been used to study and predict the complex interactions between hemodynamics and coagulation activation factors, platelets, and platelet agonists [13]. Thrombosis models for medical devices have been derived from mathematical models designed to predict in-vivo clot formation. For example, Taylor et al. [14] adapted five existing models [15–19] to develop a macroscopic-continuum thrombosis model. Another example, is the model of Wu et al. [20] who extended the model of Sorensen et al. [18] and included platelet erosion effects first presented by Goodman et al. [21]. Other models have been developed following the same principle of adapting existing complex descriptions to perform simulations of thrombogenesis in complex geometries [22–25].

In most device-related thrombosis models, platelet activity plays a central and oftentimes exclusive role.

For example Blum et al. [22] and Wu et al. [26] developed models to primarily predict platelet activation and its effects in rotary blood pumps. One of the disadvantages of an overemphasis on platelet activity is that these models cannot capture the clot formation initiated by factor

XII adsorption which then activates the intrinsic pathway of the coagulation cascade [12, 27, 28]. Therefore, this deficit needs to be addressed to capture clotting across a wider range of medical devices, including those devices in which platelet activity plays a minor role, such as blood oxygenators, coils, venous catheters, vena cava filters, hemodialyzers, etc. [29].

In contrast to device-related thrombosis modeling, a vast amount of work has been performed to understand fibrin formation dynamics due to a blood vessel injury. The modeling approaches range from kinetic pathway simulations of fibrin gel formation with ordinary differential equations [30–32] to complex gelation models that include blood flow interactions [33–36], as highlighted in a recent review by Nelson et al. [37].

In the current work, the platelet based thrombosis model of Wu et al. [20] was augmented with a partial model of the intrinsic coagulation cascade that includes fibrin formation and contact activation. We calibrated the resulting fibrin-augmented model by simulating a backward-facing-step flow channel (BFS) that has been extensively characterized in-vitro [38, 39]. Finally, to further test the predictive capabilities of the model for a more clinically relevant geometry, we performed blood perfusion experiments through a microfluidic chamber mimicking a hollow fiber membrane oxygenator and validated the in-silico model against these experimental observations.

## Materials and methods

### Ethics statement

For the microfluidic blood perfusion experiments written consent was obtained from donors under an approved Carnegie Mellon University Institutional Review Board protocol.

In this section, the biochemical and fluid dynamic governing equations of the fibrin-augmented thrombosis model are presented. In addition, two numerical setups are introduced for the BFS and a hollow fiber bundle chamber chamber. The benchtop experimental setup and blood perfusion experiments of the hollow fiber bundle are also detailed.

### Thrombosis model

Following the approach of Wu et al. [20], the fibrin-augmented model combines the incompressible Navier-Stokes (NS) equations with a system of convection-diffusion-reaction (CDR) equations for the biochemical blood constituents involved in thrombus formation. As the clot grows, a hindrance term is included in the NS equations to impede blood flow in the regions where clot is present.

### Blood flow dynamics

The pressure and velocity fields $p$ and $\boldsymbol{u}$ are obtained by solving the equations of conservation of mass and linear momentum:

$$\nabla \cdot \mathbf{u} = 0 \tag{1}$$

$$\rho \left( \frac{\partial \mathbf{u}}{\partial t} + \mathbf{u} \cdot \nabla \mathbf{u} \right) = -\nabla p + \mu \nabla^2 \mathbf{u} - \frac{C_2}{(1-\phi)} f(\phi) \mathbf{u} \tag{2}$$

where $\mu$ is the asymptotic dynamic viscosity of blood and $\rho$ is the density. The scalar field $\phi$ represents the volume fraction of thrombus which comprises deposited platelets and fibrin. To avoid a singularity in the source term, the denominator is set as $(1 - \phi) \in [\epsilon, 1]$, where $\epsilon$ is a small number. $C_2$ is the hindrance constant introduced by Wu et al. [20], which assumes that the thrombus is composed of densely compact spherical particles (2.78 $\mu$m). $f(\phi)$ is the

hindrance function:

$$f(\phi) = \phi(1 + 6.5\phi) \tag{3}$$

that approximates interaction forces between thrombus and blood flow, and its functional form is taken from Batchelor [40].

The scalar shear rate $\dot{\gamma}$ which is used throughout the thrombosis model is computed via OpenFoam source code as [41]:

$$\dot{\gamma} = \sqrt{2Tr(symm(\nabla\mathbf{u}) \cdot symm(\nabla\mathbf{u}))} \tag{4}$$

## Biochemical components

The transport of the biochemical constituents in time and space is modeled using a system of CDR equations:

$$\frac{\partial c_i}{\partial t} + \nabla \cdot (c_i\mathbf{u}) = \nabla \cdot (D_i\nabla c_i) + r_i \tag{5}$$

where: $c_i$ is the concentration of species $i$, $D_i$ is the corresponding diffusion coefficient, $\boldsymbol{u}$ is the velocity vector field, and $r_i$ is the reaction source term that accounts for biochemical interactions.

The biochemical species considered in the model are: 1) **AP** activated platelets, 2) **RP** resting platelets, 3) **AP**$_d$ deposited activated platelets, 4) **RP**$_d$ deposited resting platelets 5) **ADP** adenosine diphosphate, 6) **TxA**$_2$ thromboxane A2, 7) **IIa** thrombin, 8) **II** prothrombin, 9) **ATIII** anti-thrombin, 10) **XII** coagulation factor XII, 11) **XIIa** activated factor XII, 12) **V** coagulation factor V, 13) **Va** activated factor V, 14) **Fg** fibrinogen, 15) **Fn** fibrin, 16) **Fg**$_d$ deposited fibrinogen, and 17) **Fn**$_d$ deposited fibrin.

**Platelet activity.** Fig 1 depicts the main constituents of the baseline model of Wu et al. [20] related to platelet activity and its interactions within the thrombosis process. Briefly, platelet deposition is modeled as follows:

(A) Resting and activated platelets can deposit to the surface following first-order reaction rates with the constants $k_{rpdB}$ and $k_{apdB}$, respectively. Resting platelets can be activated

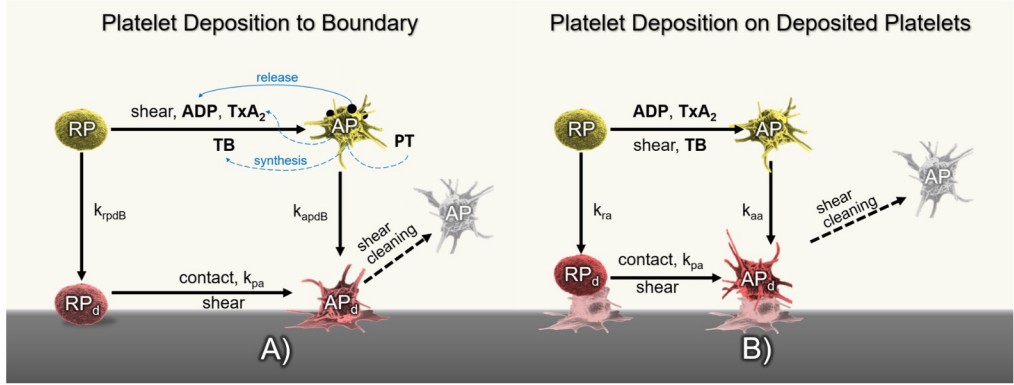

**Fig 1.** A) Diagram of platelet activation and deposition in the thrombosis model of Wu et al. [20]. $k_{rpdB}$ and $k_{apdB}$ are the rates at which resting and activated platelets deposit to the surface. Resting platelets can be activated by mechanical shear or the combination of agonists: ADP, TxA$_2$ and thrombin. Once deposited, AP$_d$ can be detached by flow shearing forces. B) Platelet deposition on deposited platelets including cleaning by large shear stresses.

**Table 1. Biochemical species involved in platelet activity.**

| Species i | Definition | Reaction term r | Diffusion coefficient m$^2$ s$^{-1}$ |
|---|---|---|---|
| **RP** | resting platelets | $-k_{apa}\,RP - k_{spa}RP - k_{rpd}\,RP + f_{emb}RP_d$ | $1.58 \times 10^{-13}$ |
| **AP** | activated platelets | $k_{apa}\,RP + k_{spa}RP + f_{emb}AP_d - k_{apd}\,AP$ | $1.58 \times 10^{-13}$ |
| **RP$_d$** | deposited resting platelets | $(1-\theta)k_{rpd}\,RP_d - k_{apa}\,RP_d - k_{spa}RP_d - f_{emb}RP_d$ | NA |
| **AP$_d$** | deposited activated platelets | $\theta k_{rpd}\,RP + k_{apd}AP + k_{apa}RP_d + k_{spa}RP_d - f_{emb}AP_d$ | NA |
| **ADP** | adenosine diphosphate | $\lambda_j(k_{apa}\,RP + k_{spa}\,RP + k_{apa}\,RP_d + k_{spa}\,RP_d + \theta k_{rpd}\,RP)$ | $2.57 \times 10^{-10}$ |
| **TxA** | thromboxane A2 | $s_{pj}\,(AP + AP_d) - k_{TxA2}\,TxA$ | $2.14 \times 10^{-10}$ |

chemically by TxA$_2$, ADP, IIa, and mechanically by flow-induced high shear stress. Once deposited, platelets can detach from the surface due to shear cleaning.

(B)  The artificial surface is covered by an initial layer of platelets followed by additional deposition of platelets mediated by the $k_{ra}$ and $k_{aa}$ constant rates. The source terms representing platelet activation, deposition, and cleaning dynamics are listed in Table 1. The constant parameters involved in platelet activity and hindrance terms are listed in Table 2.

**Coagulation cascade and fibrin/fibrinogen deposition.**   Fibrin formation and deposition is depicted in Fig 2 that depicts the following processes.

(A)  Fig 2A shows fibrinogen and fibrin depositing to a surface according to $k_{FgdB}$ and $k_{FndB}$ following the Vroman effect. Fibrin production is mediated by IIa which can be synthesized from activated platelets or via coagulation reactions [18, 46]. Coagulation reactions are modeled using the partial model of Méndez Rojano et al. [47] which considers six coagulation reactions to form thrombin. In the present model, coagulation reactions are triggered through factor XII activation at the biomaterial surface. As shown in Fig 3, the model considers an auto-activation loop through factor V and an inhibition step through antithrombin (ATIII). A fibrin formation step was introduced from the reduced order model from Chen et al. [48]. The fibrin formation reaction follows Michaelis-Menten kinetics applying the original measured kinetic parameters.

(B)  Subsequently, additional fibrin can deposit according to $k_{Fn}$ to constitute a growing thrombus comprised of both platelets and fibrin. Fibrin is also susceptible to shear clearing. (See Fig 2B.) Table 3 lists the CDR source terms for the constituents involved in fibrin formation and coagulation reactions. The constant parameters involved in coagulation reactions and fibrin dynamics are listed in Table 4.

## Numerical simulations and experimental methods

Two numerical experiments were conducted to calibrate and validate the thrombosis model. The BFS experimental configuration of Taylor et al. [38] and a microfluidic fiber bundle chamber from Lai [49] were simulated using OpenFoam [41].

**Backward facing step.**   To calibrate the thrombosis model, we simulated a backward-facing-step flow channel corresponding to experimental preparation reported by Yang et al. [39]. In this configuration, both human and bovine blood were circulated for 30 min at a flow rate of 0.76 L min$^{-1}$. Yang et al. stored the blood in citrate anti-coagulated blood bags, and prior to use, the blood was recalcified. Magnetic resonance imaging (MRI) was used to quantify

**Table 2. Thrombosis model parameters.** If units are not specified, parameters are non-dimensional. For more details on how $k_{rpd}$ and $k_{apd}$ are computed please refer to the original paper of Wu et al. [20].

| Variable | Description | Value/expression | Ref |
|---|---|---|---|
| $\rho$ | Blood density (Human) | $1050 \text{ kg m}^{-3}$ | [39] |
| $\mu$ | Asymptotic Blood Viscosity (Human) | 4.2 cP | [39] |
| $C_2$ | Particle packing constant | $2 \times 10^6 \text{ kg m}^{-3} \text{ s}^{-1}$ | [20] |
| $\phi$ | Thrombus Volume Fraction | $\frac{RP_d + AP_d}{PLT_{max}} + \frac{Fn_d + Fg_d}{Fn_{max}}$ | - |
| $PLT_{max3D}$ | Maximum platelet concentration in volume | $2.518e16 \text{ PLT m}^3$ | [20] |
| $PLT_{s,max}$ | Surface capacity of the surface for platelets | $7 \times 10^{10} \text{ PLT m}^2$ | [20] |
| $Dia_{PLT}$ | Platelet Diameter | $2.78 \times 10^{-6} \text{m}$ | [20] |
| $Fn_{max3D}$ | Maximum Fibrin concentration in volume | $18 \times 10^5 \text{ nmol m}^{-3}$ | [42] |
| $Fn_{s,2D}$ | Maximum Fibrin concentration over surface | $14.7 \text{ nmol m}^{-2}$ | [43] |
| $\theta$ | Platelet activation by contact | 1 | [20] |
| $w_{ADP}$ | Platelet activation weight by ADP | 1 | [18] |
| $w_{TxA2}$ | Platelet activation weight by TxA2 | 3.3 | [18] |
| $w_{IIa}$ | Platelet activation weight by IIa | 30 | [18] |
| $t_{ct}$ | Characteristic activation time by agonist | 1 s | [18] |
| $t_{act}$ | Characteristic activation time by shear stress | 0.1–0.5 s | [44, 45] |
| $ADP_c$ | ADP critical concentration | $2.0 \times 10^6 \text{ nmol m}^{-3}$ | [20] |
| $TxA2_c$ | TxA2 critical concentration | $0.6 \times 10^6 \text{ nmol m}^{-3}$ | [20] |
| $IIa_c$ | Thrombin critical concentration | $0.1 \times 10^{10} \text{ nomol m}^{-3} {}^*$ | [20] |
| $\lambda_j$ | Amount of agonist $j$ release per platelet | $2.4 \times 10^{-8} \text{ mol PLT}^{-1}$ | [20] |
| $s_{pj}$ | Synthesis rate constant of TxA2 | $9.5 \times 10^{-12} \text{ mol PLT}^{-3} \text{ s}^{-1}$ | [20] |
| $k_{TxA2}$ | Inhibition rate constant of TxA2 | $0.0161 \text{ s}^{-1}$ | [20] |
| $k_{T,1}$ | First Order rate constant | $13.333 \text{ s}^{-1}$ | [20] |
| $K_t$ | Dissociation constant for heparin/IIa | $3.5 \times 10^4 \text{ nmol m}^{-3}$ | [20] |
| $\phi_{rt}$ | Thrombin generation rate on the surface of RP | $6.5 \times 10^{-16} \text{ m}^3 \text{ nmol}^{-1} \text{ PLT}^{-1} \text{ Us}^{-1}$ | [20] |
| $\phi_{at}$ | Thrombin generation rate on the surface of AP | $3.69 \times 10^{-15} \text{ m}^3 \text{ nmol}^{-1} \text{ PLT}^{-1} \text{ Us}^{-1}$ | [20] |
| $k_{apa}$ | Biochemical platelet activation rate | $\begin{cases} 0, & \Omega < 1.0 \\ \frac{\Omega}{t_{ct}}, & \Omega \geq 1.0 \\ \frac{1}{t_{act}}, & \frac{\Omega}{t_{ct}} \geq \frac{1}{t_{act}} \end{cases} \text{ s}^{-1}$ | [20] |
| $\Omega$ | Agonist activation weight | $\sum_{j=1}^{n_a} w_j \frac{a_j}{a_{jcrit}}$ | [20] |
| $k_{spa}$ | Mechanical platelet activation rate | $\frac{1}{t_{ct,spa}} = \begin{cases} \frac{1}{4.0 \times 10^6 \tau^{-2.3}} & t_{ct,spa} > t_{act} \\ \frac{1}{t_{ct,sp}} & t_{ct,spa} < t_{act} \end{cases} \text{ s}^{-1}$ | [20] |
| $f_{emb}$ | Platelet embolization rate | $\left(1 - exp\left(-0.0095 \frac{\tau}{\tau_{embb}}\right)\right) \text{ s}^{-1}$ | [20] |
| $k_{ra}$ | Deposition rate constant of RP to $RP_d$ | $3.0 \times 10^{-6} \text{ m s}^{-1}$ | [20] |
| $k_{aa}$ | Deposition rate constant of AP to $AP_d$ | $3.0 \times 10^{-5} \text{ m s}^{-1}$ | [20] |
| $k_{rpd}$ | Deposition rate flux of RP, $f$ refers to the face of a mesh cell and $\vec{n}$ is the unit normal face | $div(k_{pd,f}\vec{n})k_{ra}$ | - |
| $k_{apd}$ | Deposition rate flux of AP, $f$ refers to the face of a mesh cell and $\vec{n}$ is the unit normal face | $div(k_{pd,f}\vec{n})k_{aa}$ | - |
| $k_{FnB}$ | Deposition kinetic rate of fibrinogen to surface | $5.0 \times 10^{-12} \text{ m s}^{-1}$ | (Calibrated) |
| $k_{FgB}$ | Deposition kinetic rate of fibrinogen to surface | $5.0 \times 10^{-14} \text{ m s}^{-1}$ | (Calibrated) |
| $k_{Fn}$ | Deposition of fibrin to thrombus | $2.25 \times 10^{-7} \text{ m s}^{-1}$ | (Calibrated) |
| $k_{rpdB}$ | Deposition rate of resting platelets to surface | $1.0 \times 10^{-22}$ | [20] |
| $k_{apdB}$ | Deposition rate of activated platelets to surface | $1.0 \times 10^{-7}$ | [20] |
| $\tau_{Emb}$ | Platelet shear embolization related constant | $9 \text{ dyne cm}^{-2}$ | [14] |
| $\tau_{EmbB}$ | Platelet shear embolization from surface | 0.275 | [14] |

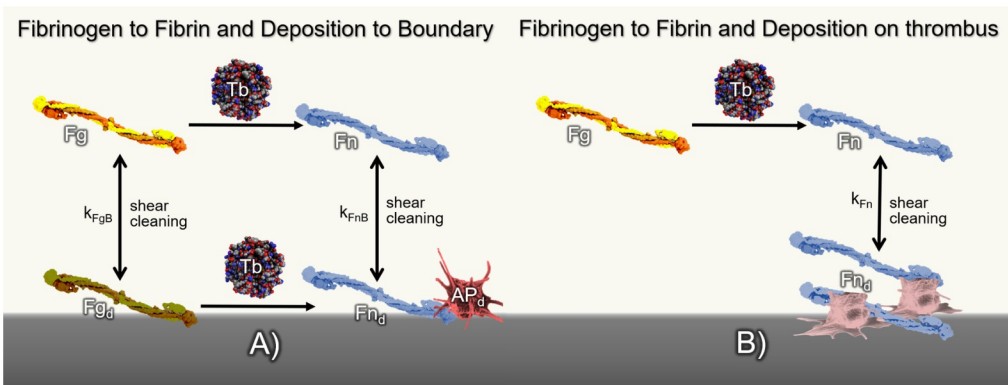

**Fig 2.** A) Schematic depiction of fibrin production due to thrombin (Tb) cleavage and deposition of fibrinogen and fibrin at $k_{FgB}$ and $k_{FnB}$ kinetic rates, respectively. Deposited fibrin and fibrinogen ($Fn_d$, $Fg_d$) detach from the boundary or thrombus due to the flow shear stress. B) Fibrin deposition to existing thrombus.

thrombus formation inside the recirculating region of the BFS in real-time. The evolution of thrombus height and length were used as benchmarks to calibrate our model.

Following the experimental protocol, two simulations were performed using human blood and bovine blood. Fig 4 shows the steady-state velocity field for the case with human blood.

A 2D approach was chosen to economize computational time and facilitate the calibration of three parameters related to fibrin and fibrinogen deposition. (See Table 2.) The outflow height of the domain was $H = 10$ mm with a step height of $h_s = 2.5$ mm. The Reynolds number in these cases was $Re_{human=411}$ and $Re_{bovine=517}$, based on the bulk velocity of $U_x = 0.23$ m s$^{-1}$, asymptotic kinematic viscosity values of $v_{human} = 4.4$ cP and $v_{bovine} = 3.5$ cP, and the inlet height of $h_{inlet} = 7.5$ mm. A fixed uniform velocity profile of $U_x = 0.23$ m s$^{-1}$ was applied at the inlet boundary. Non-slip boundary conditions (BC) were used at the walls. A zero gradient velocity BC was applied at the outlet face. For pressure BCs, a fixed total pressure value of zero was applied at the outlet, and zero gradient BCs were used at the inlet and walls. For the

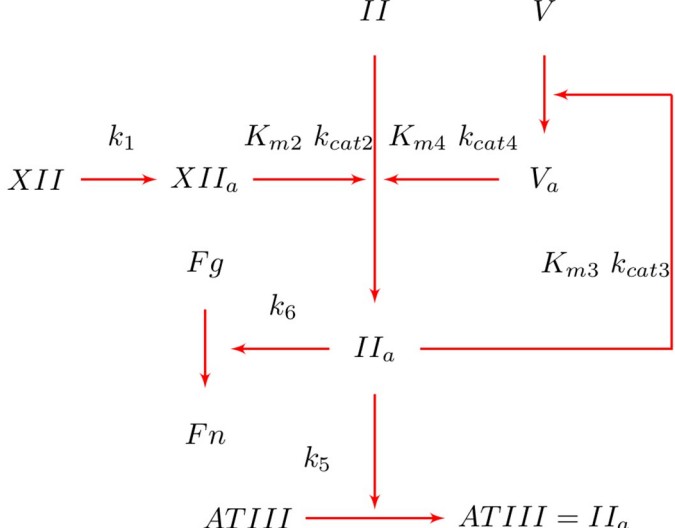

**Fig 3. Partial model of the coagulation cascade.**

**Table 3. Coagulation factors involved in the thrombosis model definition and reaction terms.**

| Species i | Definition | Reaction term r | Diffusion Coefficient m$^2$ s$^{-1}$ |
|---|---|---|---|
| XII | Factor XII | $-k_{1c}XII$ (Only applied as reactive boundary surfaces) | $3.97 \times 10^{-11}$ |
| XIIa | Activated factor XII | $k_{1c}XII$ (Only applied as reactive boundary surfaces) | $5 \times 10^{-11}$ |
| V | Factor V | $-k_{cat3}IIaV/(k_{m3} + V)$ | $3.12 \times 10^{-11}$ |
| Va | Activated factor V | $k_{cat3}IIaV/(k_{m3} + V)$ | $3.82 \times 10^{-11}$ |
| II | Prothrombin | $-\frac{k_{cat2}XIIaII}{k_{m_2}+II} - \frac{k_{cat4}VaII}{k_{m4}+II} - \varepsilon II(\phi_{at}(AP + APd) + \phi_{rt}(RP + RPd))$ | $5.21 \times 10^{-11}$ |
| IIa | Thrombin | $\frac{k_{cat2}XIIaII}{k_{m_2}+II} + \frac{k_{cat4}VaII}{k_{m4}+II} - k_{5c}IIaATH + \varepsilon II(\phi_{at} * (AP + APd) + \phi_{rt} * (RP + RPd))$ | $6.47 \times 10^{-11}$ |
| Fg | Fibrinogen | $f_{emb}Fgd - \alpha k_{cat6}FgII_a/(k_{m6} + Fg)$ | $3.1 \times 10^{-11}$ |
| Fn | Fibrin | $f_{emb}Fnd - k_{Fnd}Fn + \alpha k_{cat6}FgII_a/(k_{m6} + Fg)$ | $2.47 \times 10^{-11}$ |
| Fg$_d$ | Deposited fibrinogen | $k_{Fgd}Fg - \alpha k_{cat6}Fg_d II_a/(k_{m6} + Fg_d) - f_{emb}Fg_d$ | NA |
| Fn$_d$ | Deposited fibrin | $k_{Fgd}Fn + \alpha k_{cat6}Fg_d II_a/(k_{m6} + Fg_d) - f_{emb}Fn_d$ | NA |
| ATH | Anti-Thrombin | $-k_{5c}II_aATH$ | $5.57 \times 10^{-11}$ |

**Table 4. Kinetic constants used in the partial coagulation model.**

| Variable | Description | Value/expression | Ref |
|---|---|---|---|
| $k_{1s}$ | Surface contact activation of FXII | $6.4 \times 10^{-3}$ m s$^{-1}$ | [47] |
| $k_{m2}$ | Michaelis-Menten rate of II activation by XIIa | $8.95 \times 10^3$ nmol m$^{-3}$ | [47] |
| $k_{cat2}$ | Catalytic rate constant of II activation by XIIa | 8.21 s−1 | [47] |
| $k_{m3}$ | Michaelis-Menten rate of V activation by IIa | 2000 nmol m$^{-3}$ | [47] |
| $k_{cat3}$ | Catalytic rate of V activation by IIa | 0.0035 s−1 | [47] |
| $k_{m4}$ | Michaelis-Menten rate of II activation by Va | $8.25 \times 10^5$ nmol m$^{-3}$ | [47] |
| $k_{cat4}$ | Catalytic rate of II activation by Va | 4.98 s−1 | [47] |
| $k_5$ | Inhibition rate of IIa by ATH | $7.79 \times 10^{-9}$ nmol m$^{-3}$ s−1 | [47] |
| $k_{m6}$ | Michaelis-Menten rate of Fg to Fn by IIa | $6.5 \times 10^6$ nmol m$^{-3}$ | [48] |
| $k_{cat6}$ | Catalytic rate of Fg to Fn by IIa | 80 s−1 | [48] |
| $\alpha$ | Effectiveness factor | 0.05 | [48] |

biochemical species, the physiologic concentration values listed in Table 5 were applied at the inlet. The initial condition was set to 0.1% of the inlet concentration for each species.

The reactive boundary conditions are provided in Table 6, adopted from those published by Wu [20] and supplemented with reactive boundary conditions for factor XII and XIIa,

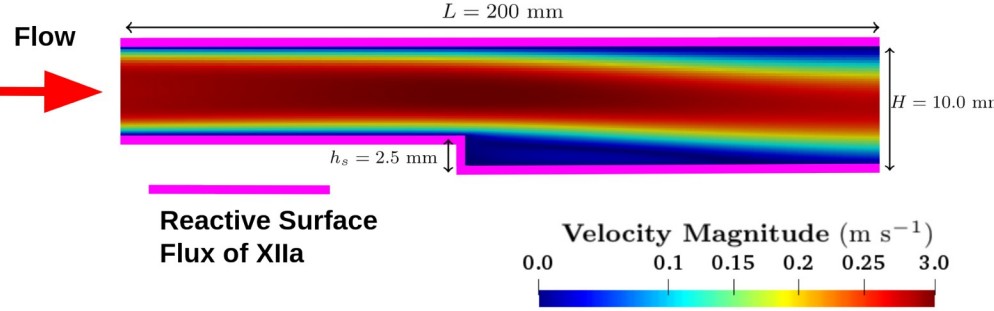

**Fig 4. 2D BFS computational domain.** Dimensions were taken from experimental setup of Taylor et al. [38]. The magenta lines denote biomaterial walls where factor XII activation boundary conditions were applied.

**Table 5. Baseline inlet concentration values.**

| Species | Value |
|---------|-------|
| AP | Human, Bovine : 1.85, 2.54 × 10$^{12}$ Plt m$^{-3}$ |
| RP | Human, Bovine : 1.85, 2.54 × 10$^{14}$ Plt m$^{-3}$ |
| PT | 933 × 10$^3$ nmol m$^{-3}$ |
| ATIII | 1665 × 10$^3$ nmol m$^{-3}$ |
| V | 6.2 × 10$^3$ nmol m−3 |
| XII | 93.6 nmol m−3 |
| Fg | 18000 × 10$^3$ nmol m−3 |

**Table 6. Reactive boundary conditions.** $S = 1 - phi$ is used to quantified available binding sites.

| Species | Boundary condition |
|---------|--------------------|
| RP | $-Sk_{rpdb}[RP] + f_{emb}[RP_d]$ |
| AP | $-Sk_{apdb}[AP] + f_{emb}[AP_d]$ |
| RP$_d$ | $\int_0^t (1-\theta)Sk_{rpdb}[RP] - k_{apa}[RP_d] - k_{spa}[RP_d] - f_{embb}[RP_d]dt$ |
| AP$_d$ | $\int_0^t Sk_{apdb}[AP] + \theta Sk_{rpdb}[RP] + k_{apa}[RP_d] + k_{spa}[RP_d] - f_{embb}[AP_d]dt$ |
| XII | $-k_{1s}[XII]/D_{XII}$ |
| XIIa | $k_{1s}[XII]/D_{XII}$ |
| II | $-\epsilon[PT](\phi_{at}[AP_d] + \phi_{rt}[RPd])$ |
| IIa | $[PT](\phi_{at}[AP_d] + \phi_{rt}[RPd])$ |
| Fg | $-Sk_{FgB}[Fg] + f_{emb}[Fg_d]$ |
| Fg$_d$ | $\int_0^t Sk_{FgB}[Fg] - f_{emb}[Fg_d] - \alpha k_{cat6}Fg_d II_a/(k_{m6} + Fg_d)dt$ |
| Fn | $-Sk_{FnB}[Fn] + f_{emb}[Fn_d]$ |
| Fn$_d$ | $\int_0^t Sk_{FnB}[Fn] - f_{emb}[Fn_d] + \alpha k_{cat6}Fg_d II_a/(k_{m6} + Fg_d)dt$ |

which in turn are modeled as a diffusive flux proportional to the kinetic constant $k_{1s}$ and inversely proportional to the diffusivity $D_{XII}$, as described in Méndez Rojano et al. [50]. In addition, boundary conditions were provided for the deposition of fibrinogen and its conversion to fibrin. The same shear cleaning term is considered for deposited platelets, fibrinogen and fibrin. (These conditions neglect the Vromann effect.) The cleaning by large shear rates for both platelets and fibrin mimics the effect of flow driven thrombus embolization and erosion. This term follows the work of Goodman et al. [21] and Wu et al. [20]. As this is the first step towards a more comprehensive model, the erosion term was considered the same for both platelets and fibrin regardless of their hydrodynamic drag.

The specific model parameters used in the study are listed in Tables 2 and 4; these values were also used for the hollow fiber bundle simulation, described in the next section. For the bovine simulation, the parameter values are listed in Table 7.

A mesh convergence study was performed using three levels of refinement and assessing the recirculation length predicted by simulations without the thrombosis model enabled.

**Table 7. Model parameters used in the bovine BFS simulation.**

| Variable | Value/expression |
|----------|------------------|
| $k_{FnB}$ | $1.0 \times 10^{-12}$ m s$^{-1}$ |
| $k_{FgB}$ | $1.0 \times 10^{-14}$ m s$^{-1}$ |
| $k_{Fn}$ | $2.15 \times 10^{-7}$ m s$^{-1}$ |

**Table 8. Grids used in the mesh convergence study for the BFS case.** The meshes are composed of uniform quadrilateral elements. The relative error is based on the predicted recirculation length.

| Grid | Elements | Element dimension (m) | Relative Error (%) |
|---|---|---|---|
| Coarse | 70,400 | $10 \times 10^{-5}$ | NA |
| Medium | 158,400 | $8 \times 10^{-5}$ | 0.21 |
| Fine | 280,000 | $6 \times 10^{-5}$ | 0.001 |

The results are listed in Table 8. The medium mesh was retained for the BFS thrombosis simulations.

An Euler scheme was used for time discretization and second order Gauss linear schemes were used for space discretization. Scalar species bounded corrections were used. A dual-time-step strategy was used in the thrombosis simulation. A large time step was used to advance the biochemical species and thrombus formation ($dt$ = 0.01 s), while the flow equations were solved with a smaller time step to ensure numerical stability ($dt$ = 0.1 $\mu$s, $CFL << 1$). This strategy has been previously used in [51, 52] showing minimal impact on the simulation results. To improve the stability of the thrombosis simulation, a steady-state flow solution was used as the initial condition. The transient PIMPLE algorithm was used to solve the pressure velocity coupling. In both cases the same time step approach and discretization schemes were used.

**Microfluidic hollow fiber bundle blood oxygenator chamber. Benchtop experiments.** A flow chamber was designed to emulate the local conditions of an artificial lung hollow fiber bundle using computer-aided design (SolidWorks, Waltham, MA) (See Fig 5A.) The microfluidic chamber modeled a small section of a clinical oxygenator. The flow chamber was 7.3 mm wide, 3 mm tall, and 20 mm long with a barbed inlet and outlet for 1/16" tubing. Hollow fibers were represented with solid rods of 380 $\mu$m in diameter, spaced evenly inside the flow chamber to create a packing density of 40%. The chamber was manufactured with a DLP 3D printer (Ember Autodesk, San Rafael, CA) for uniformity, using an acrylate resin (PR-48 Colorado Polymer Solutions, CO). After cleaning and sterilization, the chamber was incorporated into a benchtop blood flow circuit to test the effect of various fiber bundle parameters on clot formation. Briefly, experiments were conducted using human blood from consenting donors. Blood was anticoagulated with 0.1 U/mL heparin and placed in a 60 mL syringe. A syringe pump was used to pump blood through 1/16" Tygon tubing connected a flow chamber in a single pass circuit as shown in Fig 5B.

**Resistance measurement.** The circuit contained a manometer (See Fig 5B.) used to measure the differential pressure across the flow chamber over 15 minutes, every 3 minutes. The

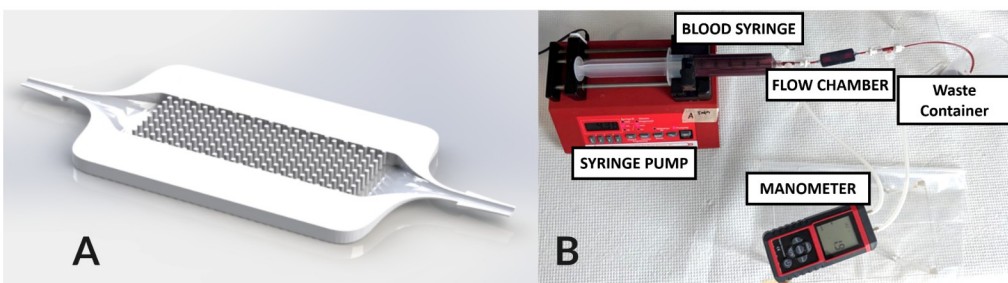

**Fig 5.** A) Microfluidic chamber geometry, upper part is hidden to visualize fibers. B) Experimental setup for hollow fiber bundle chamber experiment.

resistance across the device was measured by using $R = \Delta P/Q$, where $R$ is the resistance of the flow chamber, $P$ is the pressure drop across the flow chamber, and $Q$ is the flow rate. Baseline flow chamber resistances were measured using a glycerol-water mixture set to 3.4 cP viscosity using a viscometer (9721-B56, Cannon-Fenske). Resistances from the experiment were normalized to this baseline resistance.

**Micro-computed tomography**. Scans were used to quantify the volume of clot inside the flow chambers and to visualize the location of clot inside the chamber. After paraformaldehyde fixing of the clot, the flow chambers were scanned using a Skyscan 1172 (Bruker-Skyscan, Belgium) at the following conditions: 40 Vp, 175 $\mu$A, 300 ms exposure, 0.4 degrees rotation step, 8 frame averaging. The scans took between 4 to 6 hours. The volumes were then converted to a stack of segmented image files for analysis (FIJI, ImageJ, USA). A 3D probability map was created for the flow chamber by averaging 6 volumes from the scans.

**Staining and Imaging of Clot**. To visualize the two components of the model, a CD61 stain for platelets (Tyr773 44–876G, Thermofisher) and a stain for fibrin (F9902, Sigma-Aldrich) were used. Fluorescence microscopy was conducted on the flow chambers (A1 R+ HD25 Nikon, USA) and an image stack was acquired for 20 slices, each 5 $\mu$m apart. Fields of interest were imaged, representing the front side of the fiber facing flow, the backside of the fiber, and the center spot between two fibers.

**Thrombosis simulation**. To economize computational resources, only a quarter of the full geometry was simulated using two symmetry boundary conditions (see Fig 6A.) A fixed flow rate BC was used at the inlet with a value of $1.825 \times 10^{-8}$ m$^3$ s$^{-1}$ corresponding to 4.38 mL min$^{-1}$. No-slip BCs were applied at the fibers and walls of the channel. Other velocity and pressure BCs were identical to the BFS case.

For the biochemical species, a fixed value BC was used at the inlet using human physiologic values for the concentration of clotting factors, whereas platelet count was set to the value measured in the experiments. The values are summarized in Table 9. Reactive boundary conditions were applied at the walls of the microfluidic chamber and outer surfaces of the hollow fibers following Table 6. As mentioned previously, platelets, fibrinogen, and fibrin can deposit on reactive boundaries. In addition, factor XII activation takes place only on reactive boundaries.

Fig 6B shows a portion of the mesh that consisted of 2,756,704 elements with a boundary layer thickness resolution of 5 $\mu$m. A mesh convergence analysis was performed using the pressure drop across the microfluidic chamber (prior to thrombus growth) as the comparison metric. The mesh sizes used in this study are listed in Table 10, the relative error for the medium and fine mesh were 1.18% and 0.26%, respectively. The reported thrombosis results correspond to the medium sized mesh. A complementary mesh convergence analysis using coagulation factors scalar fields is included in the S1 Text.

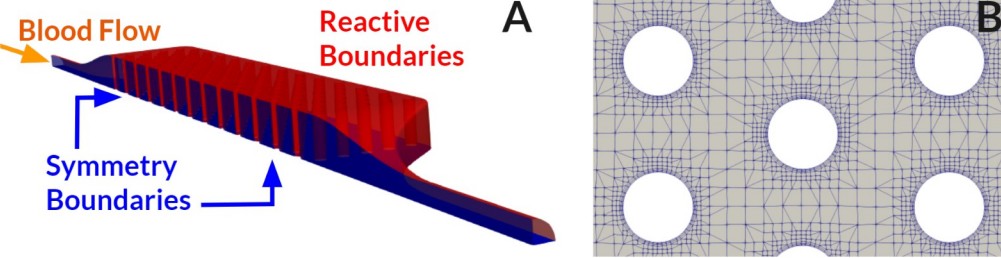

**Fig 6.** A) Simulation domain comprised a quarter of the full geometry, taking advantage of device symmetries. Boundaries colored blue were set as symmetry planes, and boundaries colored red correspond to reactive boundary conditions. B) Mesh boundary layers at hollow fibers in the microfluidic chamber.

**Table 9. Inlet concentration for biochemical species for the microfluidic chamber case.** The platelet count was taken from Lai [49].

| Species | Concentration |
|---|---|
| RP | 237 kPlt $\mu L^{-1}$ |
| AP | 2.37 kPlt $\mu L^{-1}$ |
| XII | 93.6 nmol m$^{-3}$ |
| V | $6.2 \times 10^3$ nmol m$^{-3}$ |
| II | $933 \times 10^3$ nmol m$^{-3}$ |
| ATIII | $1665 \times 10^3$ nmol m$^{-3}$ |
| Fg | $18000 \times 10^3$ nmol m$^{-3}$ |

**Table 10. Mesh sizes used in mesh convergence analysis for the microfluidic chamber case.** Element dimension size corresponds to the smaller elements located at the fiber boundary layers. The meshes are composed of hexahedral, prism and polyhedral elements. The relative error was computed using the pressure drop across the chamber.

| Grid | Elements (x10$^6$) | Element dimension (m) | Relative Error (%) |
|---|---|---|---|
| Coarse | 2.01 | $8 \times 10^{-6}$ | NA |
| Medium | 2.7 | $5 \times 10^{-6}$ | 1.18 |
| Fine | 4.2 | $2 \times 10^{-6}$ | 0.26 |

**Computational resources**. Thrombosis simulations were conducted on a workstation PC with two Intel Xeon CPU E5–2699 v3 processors @ 2.30GHz with 18 cores and 36 threads. The 2D BFS thrombosis simulation took 3 hours wall time using 32 threads while the 3D fiber bundle chamber simulation took 126 hours wall time using 72 threads.

## Results

### Backward Facing Step (BFS)

Fig 7 shows the time course of thrombus formation in the recirculating region of the BFS. The plots of the velocity magnitude reveal the effect of the growing thrombus on the flowing blood:

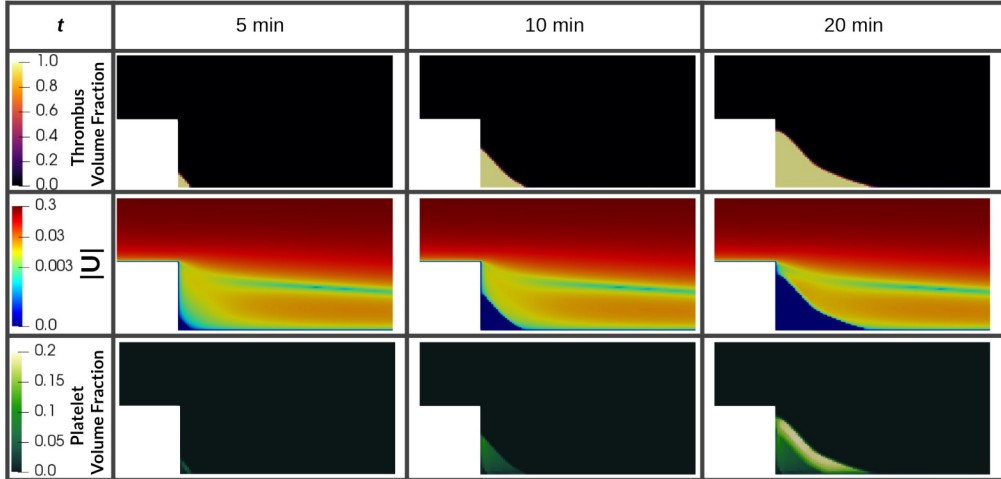

**Fig 7. Time course of scalar fields of thrombus and platelet volume fractions.** In the middle row shows the log10 scaled velocity magnitude scalar field and the influence of the growing thrombus in the flow.

                                    

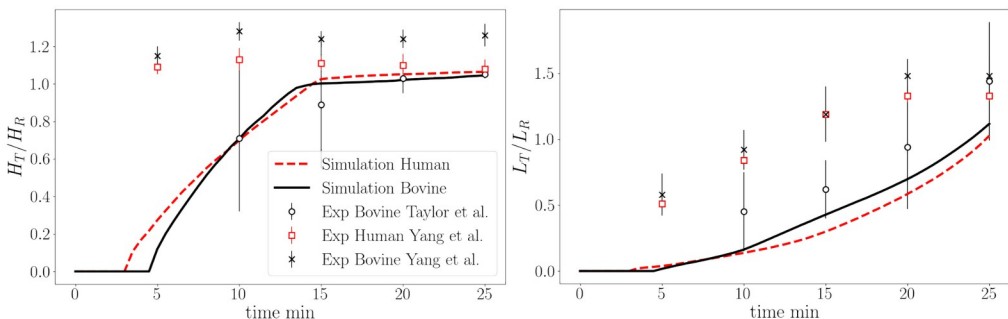

**Fig 8. Quantitative comparison of simulation results and experimental data for BFS thrombus height normalized by step height and thrombus length normalized by the initial flow recirculation length.**

the flow is arrested in the region occupied by the clot, and the topology of the recirculating region is distorted. The plots of platelet volume fraction show that platelets deposit preferentially in the outermost layer of the thrombus with a dense fibrin core. The volume fraction of platelets is low (20%) compared with fibrin which is the main constituent. A small thrombus is present at the reattachment point as in the simulations of [14] (see S1 Fig).

The normalized thrombus height and length (normalized by step height and initial recirculation length, respectively) are plotted in Fig 8 and compared with experimental data for bovine blood from Taylor et al. [38] and bovine and human data from Yang et al. [39] For both bovine and human simulations, the general thrombosis growth trend was similar, characterized by an accelerating monotonic growth in length and an asymptotic growth in height, reaching a plateau at the top of the step. The simulations underpredicted the initial growth rates compared to the MRI data reported by Yang et al., however, the total height and length approached experimental values at 25 minutes. Since the data are nondimensionalized with the recirculating zone length and height, the two simulations appear to yield the same result, but in fact the human and bovine thrombus sizes of different sizes, specifically, the human blood thrombus length and height at 25 min are slightly lower.

To demonstrate the impact of fibrin formation, coagulation cascade, and contact activation in the BFS configuration, the above results were compared to a simulation with the original platelet-based model of Wu et al. [20]. Fig 9 shows the platelet volume fraction at 10 min for the model of Wu and the thrombus volume fraction for the current thrombosis model. The kinetic rates of platelet deposition in the platelet-based model are two orders of magnitude larger than the same rates used in the present fibrin-augmented model. Despite the increased

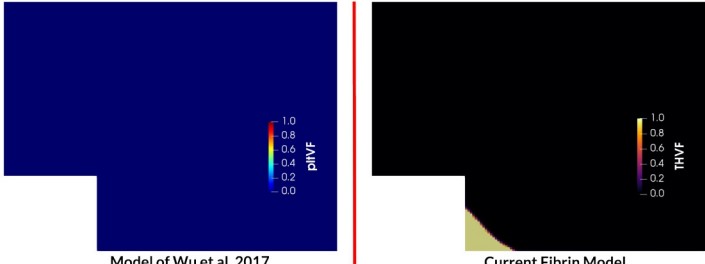

**Fig 9. Comparison of thrombus formation at 10 min for the baseline model of Wu et al. [20] in terms of platelet volume fraction (pltVF) and the current thrombosis model as the sum of fibrin volume fraction and platelet volume fraction, THVF.**

platelet deposition rates, no thrombus deposition is observed when using the platelet-based model. In contrast, the fibrin-augmented case is in line with the experiments. It should be noted that the thrombus volume fraction in our work was defined as the sum of fibrin and platelet volume fractions as contrasted with the platelet-based model that quantified thrombus only by platelet volume fraction.

### Microfluidic hollow fiber bundle chamber

Thrombus deposition in the hollow fiber bundle chamber was observed to initiate in regions of low shear rate, namely, at the lateral wall corners of the chamber and at the aft sides of hollow fibers. (See Fig 10.) These locations were also associated with low velocity magnitude and

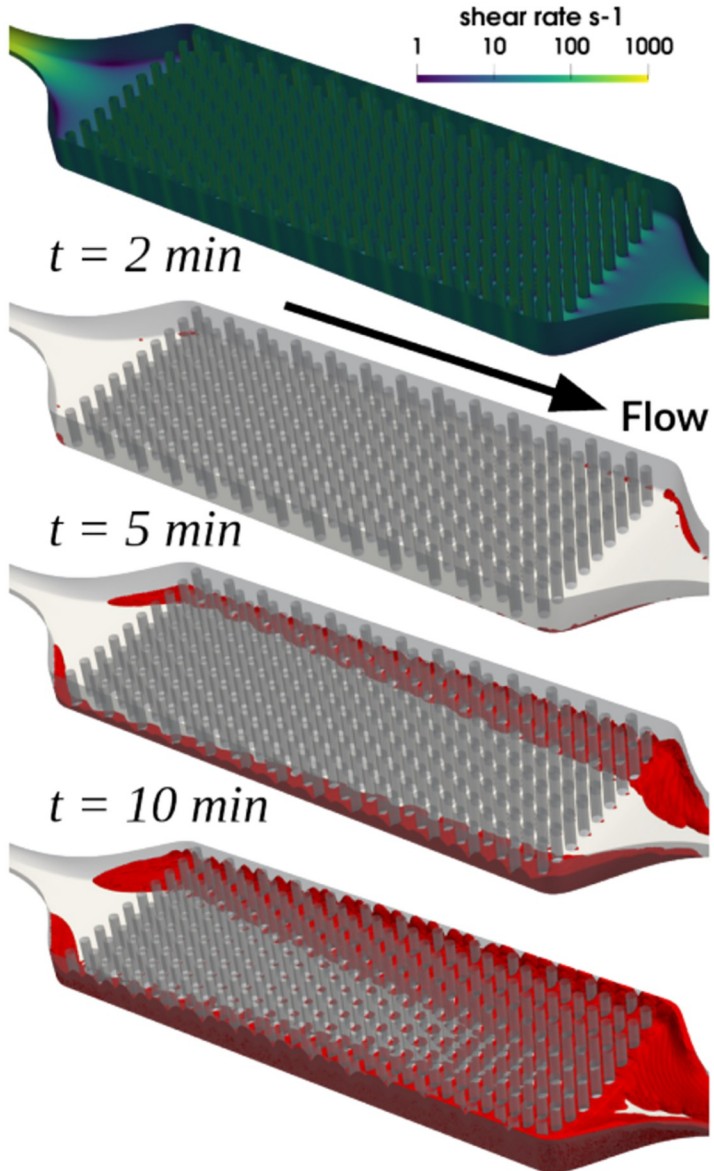

**Fig 10. Time course of thrombus formation in the hollow fiber bundle oxygenator, depicted by thrombus volume fraction threshold ($THVF > 0.1$) colored red.** The wall shear rate field ($s^{-1}$) is shown prior to any thrombus growth, i.e., $t = 0$ min.

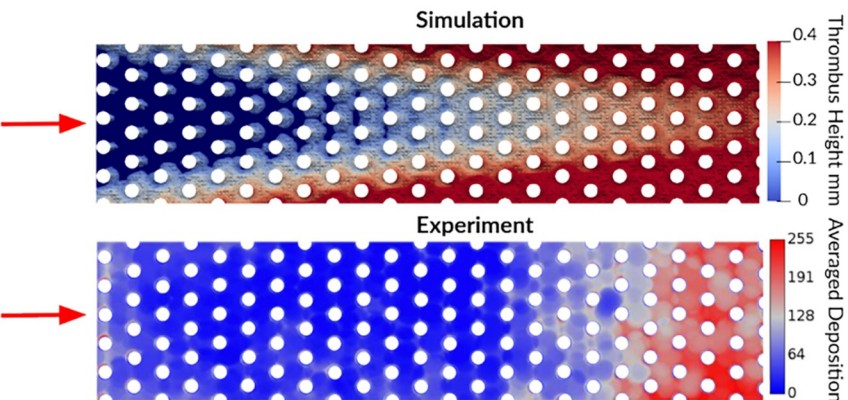

**Fig 11. Simulated and experimental thrombus formation patterns in the microfluidic hollow fiber bundle chamber at 15 min.** Thrombus height was used as a surrogate for thrombus density to compare against experimental deposition maps computed from multiple $\mu$CT averaged scans from Lai to create a clot probability map [49].

prolonged residence time, which allowed coagulation reactions to take place. Clotting formed preferentially towards the outlet region of the chamber and propagated from the lateral and lower/upper walls towards the center of the chamber.

After 15 min of thrombosis simulation, clotting formed preferentially downstream towards the narrowing outlet, as shown in Fig 11. In this figure, thrombus height was used as a surrogate for thrombus density to compare against experimental deposition maps from multiple $\mu$CT scans. The simulation predicted thrombus growth that initiated from the chamber walls and converged to restrict flow at the outlet; whereas the experiment showed a uniform growth across the chamber width that abruptly initiated at a given specific length in the flow direction. It is to be noted that only the central portion of the physical chamber was imaged that excluded wall effects observed in the experiment.

Fig 12 shows the fibrin concentration field on the middle plane at 15 min. It can be observed that the largest fibrin concentration formed near the walls and increased towards the

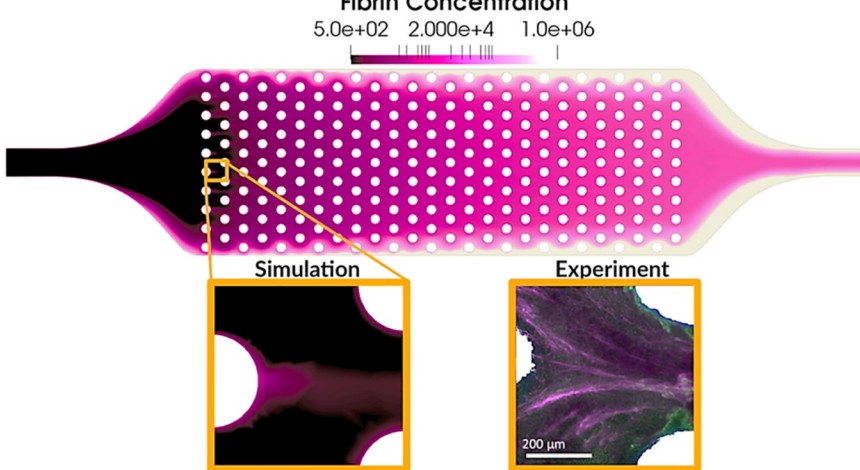

**Fig 12. Fibrin concentration field on the middle plane at 15 min.** Insets: local fibrin structure around individual hollow fibers compared against experimental images from Lai [49].

outlet. At a finer scale, the medium mesh-size model loosely captured the fibrin streak structures observed experimentally downstream of the individual hollow fibers. (See Fig 12 insets.)

## Discussion

Thrombosis in low shear conditions is characterized by a predominance of fibrin, which is the result of factor XII production stimulating the coagulation cascade [13, 53]. The current work highlights the importance of considering this mechanism in devices that have large blood-contacting surface to volume ratios, that operate at low shear rate conditions ($< 1000 \text{ s}^{-1}$), and that are characterized by low flow velocities and large residence times. A fibrin-augmented thrombosis model was introduced by extending an existing platelet-based model with key elements of the intrinsic coagulation cascade. The model was then calibrated to match thrombus length and height of published backward facing step experiments. To showcase the versatility of the model, thrombosis computations were performed for a more clinically-relevant hollow fiber bundle microfluidic chamber.

In the BFS configuration, the baseline platelet-based thrombosis model without fibrin formation and contact activation was not able to generate clotting in the recirculating region nor was it able to accurately represent the predominance of fibrin in the low-shear zone, as reported in the experimental results of Taylor et al. [38] The present BFS results illustrate the critical role of fibrin for predicting the time course of thrombus deposition. Although the BFS geometry has been commonly used by others to validate various thrombosis models for steady-state flows [23, 54–56], these models all fail to accurately characterize the time evolution of clot initiation and growth.

In contrast, the current model predictions are time-accurate and in line with the nature of thrombus constitution in low flow, low shear stress environments. In general, when thrombosis models do not account for factor XII activation, thrombus formation is typically initiated by forcing an arbitrarily high constant concentration of activated coagulation factors at user-selected specific regions of the domain. In our method, all blood-wetted walls are set to be reactive, thus allowing factor XII to be activated at these biomaterial surfaces. This feature allows flow transport to naturally react at regions where thrombi can form without the user prescribing arbitrary boundary concentrations.

A qualitative comparison of the fibrin concentration field and thrombus deposition patterns was performed in the microfluidic hollow fiber bundle chamber. Although these comparisons are only qualitative, to the extent of our knowledge, this is one of the first device-related thrombosis models that performs comparisons of scalar concentration fields with experimental observations. In addition, our results demonstrate the importance of considering fibrin as a thrombus constituent; in this regard platelet-based models would not be consistent with the thrombus composition. More qualitative comparisons are needed to fine tune model parameters.

Computational fluid dynamics (CFD) simulations of blood oxygenators have been used to identify regions prone to thrombogenesis [57, 58]. For example, Conway et al. [58] reported that a cumulative residence time along the flow axis computed from CFD simulations was partially predictive of clot burden. In a similar manner, Gartner et al. [57] used 2D CFD simulations within blood oxygenators to minimize regions of low blood velocity regions suspected of producing potential thrombosis. While this general approach is amenable to performing geometric optimization, the criteria that defines low velocity and residence times are arbitrary and ad hoc. In contradistinction, our approach is significantly more versatile and robust for predicting thrombotic initiation and deposition without a priori knowledge and can be used to extract valuable time evolution information about the thrombogenic process.

Another advantage of the present model is its amenability to integrate commonly used anti-coagulants such as Heparin, Rivaroxaban, Bivalirudin, etc. Consequently, the current model could be used to inform anticoagulant therapies in patients treated with medical devices, such as ECMO circuits and oxygenators in the context of COVID-19 where coagulopathies suggest increased thrombotic complications [59–64]. An additional future application of the current fibrin model could be to simulate fibrinolysis strategies. As highlighted by He et al. [65], another important challenge in blood-contacting medical devices is the controlled formation of neointimal tissue while preventing abnormal fibrovascular tissue known as pannus [9]. The model can be used to provide the fibrin bed field required for neointimal tissue simulations.

Several limitations are acknowledged in the current work. Although our model was able to reproduce certain time courses of thrombus growth, it predicted a slower growth rates and different deposition dynamics compared to experiments. This could be explained by the lack of accounting for red blood cells (RBC) in the model of thrombosis, which in reality are trapped within fibrin clots and significantly increases its volume and decreases its flow permeability. Another disparity was the prediction of thrombus formation at the flow reattachment point of the BFS, which, although is in line with the model of Taylor et al. [14], was not observed experimentally. A possible explanation for this is that in the experiment, contrary to the simulation, the reattachment region is not spatially fixed, but varies its location enough to promote shear-induced washout of agonists. In addition, the resistance computed from the simulation over-predicts the experimental observations: 3.7 $\frac{mmHg}{mL/min}$ versus 0.7 $\frac{mmHg}{mL/min}$ at 10 min, respectively. This can be explained by the excessive thrombus growth predicted at the outlet barb of the device, which in reality is continuously embolizing towards the waste container. A portion of the disparity could also be attributed to the clot permeability term $C_2$ adopted from Wu et al. [20] which might not be representative of red clots found experimentally. In future work, the hindrance function term can be expressed as a function of fibrin-platelet ratio to reflect the complex clot composition. The list of parameters currently included in the model is not necessarily universal or independent, and hence should be more carefully and extensively studied coupled with experimental validation as in our previous sensitivity analysis and uncertainty quantification study [52]. We have assumed that phospholipids and calcium are in excess which translates to an overestimation of thrombin and fibrin production. Additionally, the activation role of polyphosphate in the intrinsic pathway was not considered [66]. The influence of RBCs in thrombus composition, near-wall platelet margination, and non-Newtonian viscosity effects were neglected as their inclusion are anticipated to be computationally prohibitive. Another omission from the current model is any consideration of sudden embolization of clots which may overestimate the final thrombus volume. Approaches for simulating embolization such as the one presented by Zheng et al. [67] could be considered in the future. Fibrin polymerization and subsequent fibrotic structure formation were not explicitly modeled and should be considered if sparse fiber structures were to be simulated in more detail. Finally, fibrin mechanics, deformation, and contraction and its interaction with platelets and other cells [68] were not considered as this would render the simulations excessively computationally intensive.

## Conclusion

This report introduced an enhanced thrombosis model that incorporates a portion of the coagulation cascade, specifically, the contact activation system and fibrin formation. The importance of these mechanisms was illustrated in the simulation of thrombus deposition in a micro-scale hollow fiber oxygenator. The current framework can be a useful tool to predict thrombosis in medical devices operating at low shear stress conditions, and hence improve their design and inform anti-coagulation therapy and patient management.

## Supporting information

**S1 Text. Mesh convergence analysis.** A spatial grid analysis on the microfluidic fiber bundle using the concentration scalar fields of two coagulation factors is presented.
(DOCX)

**S1 Fig. Thrombus growth in backward facing step reattachment point.**
(PDF)

## Author Contributions

**Conceptualization:** Rodrigo Méndez Rojano, Angela Lai, Keith Cook, James F. Antaki.

**Data curation:** Rodrigo Méndez Rojano, Angela Lai.

**Formal analysis:** Rodrigo Méndez Rojano, Angela Lai, Mansur Zhussupbekov, Greg W. Burgreen.

**Funding acquisition:** James F. Antaki.

**Investigation:** Rodrigo Méndez Rojano, Angela Lai, Keith Cook.

**Methodology:** Rodrigo Méndez Rojano, Angela Lai, Mansur Zhussupbekov, Greg W. Burgreen.

**Project administration:** Keith Cook, James F. Antaki.

**Resources:** Keith Cook, James F. Antaki.

**Software:** Rodrigo Méndez Rojano, Mansur Zhussupbekov, Greg W. Burgreen.

**Supervision:** Greg W. Burgreen, Keith Cook, James F. Antaki.

**Validation:** Rodrigo Méndez Rojano.

**Visualization:** Rodrigo Méndez Rojano.

**Writing – original draft:** Rodrigo Méndez Rojano, Angela Lai.

**Writing – review & editing:** Rodrigo Méndez Rojano, Angela Lai, Mansur Zhussupbekov, Greg W. Burgreen, Keith Cook, James F. Antaki.

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
