## [Decision Letter · Decision Letter 0]

6 Jul 2022

Dear Ph.D. Méndez Rojano,

Thank you very much for submitting your manuscript "A fibrin enhanced thrombosis model for medical devices operating at low shear regimes or large surface areas" for consideration at PLOS Computational Biology.

As with all papers reviewed by the journal, your manuscript was reviewed by members of the editorial board and by several independent reviewers. In light of the reviews (below this email), we would like to invite the resubmission of a significantly-revised version that takes into account the reviewers' comments.

We cannot make any decision about publication until we have seen the revised manuscript and your response to the reviewers' comments. Your revised manuscript is also likely to be sent to reviewers for further evaluation.

Sincerely,

Alison L. Marsden

Associate Editor

PLOS Computational Biology

Daniel Beard

Deputy Editor

PLOS Computational Biology

Reviewer's Responses to Questions

**Comments to the Authors:**

Reviewer #1: This work by Méndez Rojano and colleagues presents a new thrombosis model developed to simulate clot formation in biomedical devices. To my knowledge, this is one of the most comprehensive thrombosis models currently available in the literature, as it includes platelet activation, platelet and fibrin deposition by low shear and through contact with reactive surfaces, also explicitly modelling (at least in part) the biochemical reactions of coagulation cascade, and allowing to predict thrombus composition. The model is then tested against the canonical backward facing step setup and against a hollow fiber membrane oxygenator, for which experimental results were available.

The authors effort to provide a comprehensive description and validation of the model is commendable. Despite failing to accurately capture some clot features observed in the experiments, the model itself is extremely promising and worth publishing at the current stage (the authors have addressed most of the limitations in the work discussion).

Major comments:

• Data and Code Availability: from PLOS CB code sharing Policy “PLOS Computational Biology requires authors to make all author-generated code directly related to their study’s findings publicly available without access restriction at the time of publication unless specific legal or ethical restrictions prohibit public sharing of code”. The authors shared their result files and post-processing codes. However, the thrombosis code is only available upon request, which constitutes an access restriction. Please make the thrombosis code and OpenFoam setup files freely available.

• Please double check consistency in variable naming and table numbers. This has made the work quite cumbersome to read. E.g., variables reported in Table 1 are not present (e.g., Spj, fstb) in Table 2 or might have changed names (e.g., krpd= krpdB?). Lines 156-160: Tables seem to be wrongly numbered. E.g. lines 157-158, the text refers to Table 6 for model parameters, but table 6 only reports inlet concentrations.

• Please provide more details about: i) your modelling choices for reactive boundary conditions (Table 7); ii) cleaning by large shear for platelet and fibrin; iii) the fibrin formation step (line 119)

• In the experiment, resistance across the device was measured, this can also be measured from the simulation results: how do the two compare?

• Meshes: please report details regarding element dimensions in Table 8 and 10. Also, please specify the element type. As for the chosen parameters (recirculation length and pressure drop) for the mesh sensitivity tests, tests conducted using species concentrations (e.g., for the species with higher and lower D) would be more appropriate. I wonder if this might also be an additional reason why predictions are not perfectly in line with experimental results.

• Numerical setup - some details about the numerical setup are not reported: discretisation schemes, pressure-volume coupling scheme. Why was such a low time step (and CFL number) required to achieve numerical stability? Was the same time step approach used in both test cases?

Minor comments:

• Please report in the results or discussion the employed computational resources and simulation time. Thrombosis models tend to be quite computationally expensive and these information would help understand how close to actual R&D application the model is and to discuss future developments to make this possible.

• Does the platelet to fibrin ratio influence thrombus resistance to flow?

• For future work, you might want to conduct a sensitivity test to better understand how choice of parameters influences your result. Could this also lead to a reduction in the number of equations?

• Fig 2, caption: Fgd and Fnd are mentioned in the wrong order, I believe, creating some confusion.

• Table 6 is mentioned in the text before Table 5

• Micro-computed tomography: please specify how long the experiment lasted

• Fig 6: no caption for part B

• Could the authors discuss why it is important to get the thrombus time evolution right (and not just the end-result)?

Reviewer #2: Mendez Rojano et al. A fibrin enhanced thrombosis model for medical devices operating at low shear regimes or large surface areas

The manuscript continues the development of a computational thrombosis model with particular emphasis on fibrin deposition in low shear/large surface areas. The manuscript is well-written. The model is compared to existing published data from a backward facing step geometry and a microfluidic oxygenator. The model is based on a previous model and modified to be more sensitive to fibrin deposition.

Generally speaking, the model compares reasonably to the experimental data. Since the model is focused on the fibrin, only Figure 12 tests this aspect of the model in a rather qualitative manner. There is not a substantial amount of rigor associated with the validation but just one gross comparison that yielded similar results.

The Discussion section also does not really describe the fibrin-centric aspects with respect to the fibrin data presented. There is no quantitative analysis or comparison to demonstrate the overall accuracy of the model.

**Have the authors made all data and (if applicable) computational code underlying the findings in their manuscript fully available?**

Reviewer #1: **No: **The authors shared their result files and post-processing codes. However, the thrombosis code and OpenFoam setup files are only available upon request, which constitutes an access restriction.

Reviewer #2: Yes

PLOS authors have the option to publish the peer review history of their article (what does this mean?). If published, this will include your full peer review and any attached files.

Reviewer #1: No

Reviewer #2: No
---

## [Decision Letter · Decision Letter 1]

15 Sep 2022

Dear Ph.D. Méndez Rojano,

We are pleased to inform you that your manuscript 'A fibrin enhanced thrombosis model for medical devices operating at low shear regimes or large surface areas' has been provisionally accepted for publication in PLOS Computational Biology.

Best regards,

Alison L. Marsden

Academic Editor

PLOS Computational Biology

Daniel Beard

Section Editor

PLOS Computational Biology

Reviewer's Responses to Questions

**Comments to the Authors:**

Reviewer #1: The authors have carefully addressed my comments and I am satisfied with the final version of the manuscript.

I did not notice any more inconsistencies in the manuscript. However, I would strongly recommend that the authors conduct a final check of the equations before the manuscript is released, to ensure these are in line with the equations used in the latest model and do not contain additional typos.

Reviewer #2: The authors have addressed my concerns.

**Have the authors made all data and (if applicable) computational code underlying the findings in their manuscript fully available?**

Reviewer #1: Yes

Reviewer #2: Yes

PLOS authors have the option to publish the peer review history of their article (what does this mean?). If published, this will include your full peer review and any attached files.

Reviewer #1: No

Reviewer #2: No

---

## [Editor Report · Acceptance letter]

27 Sep 2022

PCOMPBIOL-D-22-00848R1 

A fibrin enhanced thrombosis model for medical devices operating at low shear regimes or large surface areas

Dear Dr Méndez Rojano,

I am pleased to inform you that your manuscript has been formally accepted for publication in PLOS Computational Biology. Your manuscript is now with our production department and you will be notified of the publication date in due course.

With kind regards,

Zsofia Freund
